# Forecasting the Wear of Operating Parts in an Abrasive Soil Mass Using the Holm-Archard Model

**DOI:** 10.3390/ma12132180

**Published:** 2019-07-07

**Authors:** Jerzy Napiórkowski, Magdalena Lemecha, Łukasz Konat

**Affiliations:** 1Chair of Vehicles and Machinery Exploitation, The Faculty of Technical Sciences, The University of Warmia and Mazury in Olsztyn, M. Oczapowskiego 11, 10-719 Olsztyn, Poland; 2Faculty of Mechanical Engineering, Department of Materials Science, Strength and Welding, Wroclaw University of Technology, Smoluchowskiego 25, 50-371 Wrocław, Poland

**Keywords:** wear steel, Holm-Archard model, soil mass, forecasting the wear

## Abstract

This paper presents the forecasting of the wear of working elements in an abrasive soil mass using the theoretical wear model. One of the widely used models providing a basis for the relationships describing wear is the Holm-Archard model. This relationship describes abrasive wear because of the contact between two bodies. The model assumes that the wear of an operating part is directly proportional to the sliding force and distance and inversely proportional to the hardness of the material of the part. To date, the model has not been verified in the wear of a soil mass, which is a discrete friction surface. Four grades of steel resistant to abrasive wear, intended for the manufacturing of operating parts exposed to wear within a soil mass, Hardox 500, XAR 600, TBL Plus and B27, were subjected to testing. TBL Plus steel was characterised by the smallest wear irrespective of the soil type. In turn, the highest values of the wear were noted in the light soil for Hardox 500, in the medium soil for XAR 600, while in the heavy soil for B27. Based on the obtained results, a high correlation coefficient was noted, with the highest values obtained for light and heavy soils.

## 1. Introduction

Forecasting and mathematical description of the wear of operating parts is a complex issue due to the presence of many variables of high randomness that affect the range and intensity of its course. The interpretation of forcing on an operating part should enable the identification and selection of stable and sensitive relationships and those providing information. The variety of factors affecting the course of the wear of an operating part prevents all of them from being included in one model. There is a possibility for introducing assumptions in the research to identify significant forcings and characterise their effects on the wear process by taking into account the other effects at a constant level [1,2,3]. The basic forms of the wear include micro-cracking, micro-ploughing, and micro-fatigue. These processes are associated with the mechanical impact of abrasive material on the surface layer of the material. Depending on the type of an operating part, including the impact of the pressure exerted by the abrasive material, chemical processes associated with the impact of soil environment can occur as well [4,5]. In order to define and describe the wear phenomenon in addition to experimental testing, many mathematical models enabling the forecasting of the service life of parts were generated [1,6]. The operating parts of the tools processing an abrasive soil mass is characterised by high resistance to both wear and impact strength. This is due to the conditions of their use under which they are subjected to both the wear impact of the soil and to overloads caused by the impact on stones fixed in the soil [3]. The current state of knowledge on the wear impact of a soil mass on operating parts is insufficient to develop the principles of selecting the design and technological solutions for operating parts used for the processing of variable soil types. Understanding and describing the wear within a soil mass consists in a comprehensive description of the friction process involving the soil—an operating part—working process parameters. Particular importance is attributed to the impact of a soil mass, since soil factors mainly affect the nature of wear while material factors affect the rate of wear. The process of an operating part’s wear in a soil mass is classified as a natural process. Another disregarded factor is the specificity of abrasion in the soil, which includes:-The impact on the material of not only mineral grains but also of the organic mass whose properties define the soil as the wear-causing mass;-Disordered distribution of a very large number of abrasive grains on the active surface of the soil:-Brittleness of abrasion surfaces;-Varying shapes of abrasive grains;-Various heights of sharp edges of abrasive grains on the active surface of the soil;-The possibility of free movement of abrasive grains depending on the soil condition;-Specific chemical properties of an abrasive mass.

The literature shows that sandy soils, although composed mostly of SiO_2_, may prove to be a soft soil, since these grains are not bonded together, which increases the rate of wear compared to loamy soils [7]. Known models that describe the wear process do not explain it comprehensively and define the process partially without taking into account all parameters occurring in a tribological pair [8]. In the case of abrasive wear in two bodies to predict the wear of wear forecasting, many models have been developed, the use of which is possible in the case of simple wear by means of micro-scratching and furrowing. Among them, one of the most frequently used is the model proposed at work [9].

In this model, the loss of the element’s volume because of wear depends on the friction path, the normal load acting on the element, the hardness of the material surface and the angle of attack consuming the particles.

Studies have shown [10] that the wear rate of an operating part within the soil is affected by the same parameters that affect the development of friction force between the operating part and the soil, i.e., the type of material, processing rate, grain-size distribution and moisture content. The lack of a quantitative relationship between the friction force values and the rate of wear results from the specificity of soil impact. An increase in friction force is caused by the action of both mineral and organic compounds. However, in the case of wear, its rate is primarily determined by mineral compounds, and only after a longer period of time by organic compounds [11].

A significant number of characteristics describing the properties of an abrasive soil mass in the context of wear impact shows, on the one hand, the complexity of the phenomena under consideration and, on the other hand, causes many difficulties in the wear process analysis. In order to solve the problem of forecasting the wear in a soil mass, mathematical models are sought that would combine the intensity of wear with the soil properties, cutting parameters and properties of the material, and explain the relationships occurring between these parameters. The literature includes attempts to develop such models, but they have so far been characterised by unsatisfactory explanatory functions and practicability. The issue of modelling is a reflection of knowledge of the phenomena under study. Two groups of models can be distinguished. The first group includes intuitive models, an understanding of which enables the selection of significant parameters of the wear process in qualitative terms. The second group includes quantitative models, i.e., models obtained based on partial research problems and, in principle, they refer to specific soil conditions.

It is necessary to develop a predictive model of the wear of tools operated in the soil environment in which the replacement of various machinery components affects the costs and work schedule. Such a model can be applied during the planning and operation phases to ensure a realistic estimation of the equipment wear [12].

A commonly known model describing the wear of operating parts in the soil is the model proposed by Tenenbaum [13]. This model takes into account, inter alia, a component of axial forces, friction force and rubbing speed, with all of them exerting their effects simultaneously. On the other hand, the relationships occurring between each other are the processes of micro-cutting, multi-deformation, wear-induced disturbance of the material structure, heating, oxidation and hydrating. A different approach was proposed by Napiórkowski [2] and Owsiak [14]. In order to determine the contribution of the abrasive soil mass to the wear in their relationships, they took into account not only the parameters describing the tested material but also the soil characteristics. Another model describing the relationships between soil particles was developed by Krieg [15]. This relationship combines the finite element methods with hydrodynamic impact of particles. On its basis, local surfaces in the locations where the direction of the abrasive mass flow changes are defined. A model describing the wear of operating parts processing an abrasive soil mass during field tests was also presented in a study by Kostencki [7]. It describes the linear wear intensity defined as an absolute quotient of the loss of a material’s thickness and its friction distance.

A common model providing a basis for the relationships describing wear is the Holm-Archard model [16]. This relationship describes abrasive wear due to the contact between two bodies. The model assumes that the wear of an operating part is directly proportional to the sliding force and distance and inversely proportional to the hardness of the material of the part. Its correctness was verified during the wear with fixed abrasive grains, in which micro-cutting processes are dominant [7]. To date, the model has not been verified during the wear in a soil mass, which is a discrete friction surface. The soil is a complex object of nature characterised by specific morphological, physical, chemical and biological features. In addition to the impact of mineral grains, the friction process is also affected by the organic mass whose properties are determined by the natural environment.

The analysis of literature data shows that the existing models of wear of working elements in real soil conditions have focused on partial research problems of the wear process, most often on determining the average durability of the working element under given extortions [17,18].

The aim of the study is to assess the suitability of the Holm-Archard model for the forecasting of the wear of operating parts processing an abrasive soil mass.

## 2. Material and Methods

### 2.1. Test Material

Micro-alloyed steels with boron are of great significance in the practical use for parts processing a soil mass. The currently produced steels of this type are characterised by the addition of boron in an amount of 0.002–0.005% w/w. In this range of concentrations, boron dissolves in austenite; consequently, even with ordinary volume hardening, a homogeneous structure of fine-grained perlite or martensite with highly fragmented grains over the entire section of the part can be obtained [18]. The currently obtained martensitic structures in low- and medium-carbon steels require no tempering treatment after hardening and at the same time, very high rates of static strength and yield point and a high capacity to absorb dynamic loads can be obtained. These steels are supplied in various states of heat treatment and are characterised by very diverse morphology of the microstructure.

Due to high utility, steel is still the basic construction material used to produce operating parts of tools and selected parts of machinery. These results, on the one hand, from the advantageous ratio of manufacturing costs and, on the other, from their versatility, susceptibility to machining, weldability as well as enhanced mechanical properties. Therefore, micro-alloyed steels with boron take on great significance, particularly in areas of intensive abrasive wear.

Four grades of steel resistant to abrasive wear, intended for the manufacturing of operating parts exposed to wear within a soil mass, i.e., B27, Hardox 500, XAR 600 and TBL Plus, were subjected to testing. These steels are referred to as fine-grained constructional steels resistant to abrasive wear, in which nitrogen is bound in the form of nitrides through the addition of aluminium, and they contain an additional niobium or titanium.

B27 steel samples were collected from 10-mm-thick sheets in hardened conditions. B27 steel is available in a wide range of semi-finished products, and most frequently used in the form of both thin and thick sheets, in a series with a thickness ranging from 2.5 to 13 mm and from 5 to 80 mm. B27 steel is characterised by a fine dispersive structure of temper sorbite (high-temperature tempered martensite) (Figure 1a). As a result of low-temperature tempering, ε carbides (Fe_2_, 4C) are cohesive with the matrix. Their release reduces the carbon concentration in martensite. Tempering in the second stage, which proceeds at a temperature of 200–300 °C, transforms residual austenite into tempered martensite in a process similar to the bainitic transformation. In the next stage of tempering, the particles of this phase become coagulated, which is due to both an increase in cementite particle size and the dissolution of small particles.

Hardox 500 (Figure 1b) steel is supplied by the manufacturer in a heat-treated condition (water hardening and tempering at a temperature of 200–700 °C). Available semi-finished products of this material include thin sheets with a thickness of 2.0–6.5 mm, and thick sheets in a series with a thickness ranging from 4 to 32 mm and from 32 to 103 mm. For the purposes of testing reported in this paper, Hardox 500 steel samples were collected from 12-mm-thick sheets.

XAR 600 (Figure 1c) steel is supplied in the form of sheets with a thickness of 4–50 mm, in many heat treatment variants, including inter alia annealing (hardness of ≤ 300 HB) and water hardening (hardness of > 550 HB). The heat treatment method is determined by the manufacturer of this steel depending on the chemical composition and sheet thickness. For the testing, XAR 600 steel samples were collected from 10-mm-thick sheets supplied by the manufacturer in a hardened condition.

TBL PLUS (Figure 1d) steel is available as fine-grained constructional steel. Similar to the previously mentioned steels, nitrogen is bound in it the form of nitrides. It is supplied in the normalising state, or after normalising rolling. Steels of the B27 grade are supplied as hot rolled (hardness of 170 HB), and then hardened in water.

Measurements of the hardness of the tested materials were carried out following the Vickers method in accordance with standard PN-EN ISO 6507-1; 1999, using a semi-automatic Vickers HV10D Sinowon hardness tester. The average value of the hardness of the tested materials are as follows: B27 steel—549 HV10, TBL Plus steel—578 HV10, Hardox 500 steel—562 HV10, XAR 600 steel—637 HV10. Analyses of the chemical composition were carried in accordance with the spectral method using a GDS500A Glow Discharge Atomic Emission Spectrometer manufactured by Leco (St. Joseph, MI, USA) with the following parameters applied: U = 1250 V, I = 45 mA, argon. The obtained results were the arithmetic mean from five measurements (Table 1). For the observations of the microstructure, a Nikon Eclipse MA200 and Epiphot 200 optical microscopes were used. Images of the microstructure were recorded using a Nikon DS-Fi2 digital camera with NIS Elements software. Observations of the microstructure at higher magnifications and microanalyses of the chemical composition, morphology and phase types were carried out using a JEOL JSM-5800LV scanning electron microscope coupled with an Oxford LINK ISIS-300 energy-dispersive X-ray spectrometer.

### 2.2. Description of Tests

Samples of the analysed steels were taken in the form of rectangles with dimensions of 38 mm × 35 mm × 10 mm using methods ensuring the stability of their structure. The samples were cut out using the high-energy abrasive water jet. The finishing of the samples to the required surface roughness was carried out on a flat-surface grinder in order to obtain roughness within the range of Ra 0.30–0.46 μm. The samples were placed in specially prepared holders which were fixed in the cultivator teeth (Figure 2). The tests were carried out in six replications.

For each material, the average value of the results obtained was determined, whereas the sample spread was determined by means of standard deviation.

## 3. Theory

### 3.1. Assumptions for the Model

Data for the model were collected during the processing of a soil mass using an agricultural tractor with a cultivator with a depth of cut of 0.1 m. The average speed of the unit was approximately 1.9 m/s. The total friction distance was 10,000 m. The samples were weighed every 2000 m.

The tests were carried out under natural operating conditions, on a stubble field, in three types of soils: loamy sand (light soil), light loam (medium soil) and common loam (heavy soil). The grain-size distribution testing was carried out using the laser diffraction method using a Mastersizer 2000 laser particle-size analyser (Malvern Panalytical Support and Services, United Kingdom) in accordance with the standard ISO 13320 (Table 2).

The soils were in a moist condition with a water content of 11–12% for the light soil, and 14–16% for the medium and heavy soils, respectively.

Given the similar physical properties (similar specific gravity values) of the tested materials in order to determine the loss, the focus was set on the mass wear which was determined using Equation (1).

The testing was conducted in August and September in a field with moisture content of 11–16%. Mass wear was determined based on the following formula:(1)$$Z_{m}=m_{p}-m_{k}\left(g\right)$$
where *Z_m_* is the mass wear (g), *m_p_* is the sample weight before testing (g), *m_k_* is the sample weight after testing (g).

### 3.2. Assumptions for the Model

In order to describe the rate of wear, the Holm–Archard model with the following mathematical form was applied:(2)$$Iz=\frac{k\times P\times L}{3H}$$
where *Iz* is the rate of wear, *k* is the coefficient of soil abrasive properties, *P* is the normal load, *H* is the material hardness, *L* is the friction distance.

The *k* coefficient was determined based on the mathematical form of the Holm–Archard model.

The data on the pressure force were derived from the tests carried out by the authors under the same abrasive conditions in which the tests presented in the study were performed [2] and from literature data as cited in studies by Owsiak et al. and Bernacki [19,20].

The value of the coefficient characterising abrasive properties of the soil was determined from the previously characterised empirical studies, from the following relationship:(3)$$k=\frac{3H\times I_{z}}{P\times L}$$

The pressure force (*P*) was adopted as cited in the studies by Owsiak et al. [14] and Bernacki et al. [20] who, based on empirical studies, concluded that the pressure force of a working tool tines on the soil, depending on its type, ranged from 390 to 820 N (Table 3). The acquired data concern tests were carried out for the same soil types, physical and chemical characteristics, and working process forcings in which the wear testing and analysis were carried out.

## 4. Results and Discussion

Based on the obtained images of surfaces subjected to wear (Figure 3, Figure 4 and Figure 5), it can be concluded that the wear processes, depending on the method, differed very slightly from each other irrespective of the type of steel or the grain-size distribution of soil. According to the description provided below, the wear pattern changed with the content of both dust and silt fractions.

The content of individual fractions and the interrelations between them, as a result of cohesion and adhesion forces were of fundamental importance. The surface damage descriptions depicted in the SEM micrographs show the complex nature of the soil’s wear effect. However, it should be emphasized that even in the case of homogeneous fractions, only one dominant wear mechanism is indicated. This is conditioned by the random arrangement of soil grains in relation to the processing tool. On the surfaces of the tools working on the light soil, fatigue wear processes dominate as a result of loosely bound soil grains [20]. In the subsurface layer of the material, as a result of repeated pressures, temporary and shallow changes occur, mainly consisting of the crushing of the micro-volume of the material. In the vicinity of the separated and permanent deformed material, stress concentrations occur leading to the formation of dislocations centres weakly associated with the substrate material. Local surface wrenches are the result of fatigue wear, which successively consist of elastic deformations, plastic deformations, formation of micro-volume deformations, characterized by defective structure and shearing of these micro-volumes. (Figure 3c,d). When processing these soils, wear mechanism, defined in the literature as “three body abrasion,” occurs [3,4]. In these soils, other methods characteristic of abrasive wear were also found, i.e., micro-cutting and micro-ploughing. These forms of wear rely on pressing soil particles and destroying the one-cycle surface of the material. When drawing, the material is not only separated, but also partially moved to the sides of the wear groove (Figure 4d). Micro-ploughing involves recessing of the abrasive particles into the abrasive and plastic squeezing the furrow in it, without causing separation of the material (Figure 5c) [2].

As the soil’s heaviness increased (increased clay content), ad hoc wear processes through micro-cutting began to dominate (Figure 4a). Micro-cutting dominates in soils with a significant content of skeletal parts (e.g., sand with a hardness of approximately 1200 HV), especially in soils in which various forms of silicon compounds do not have the ability to change position due to force action. This process is favoured by clay and dust fractions, which in both dry and moist soil conditions are a specific adhesive for the hard soil particles. Apart from micro-cutting, micro-ploughing has a large share in the wear in concise soils, which occurs with much higher intensity than in light soils. The phenomena of wear of the surface layer in concise soils quite well describes the “two-body abrasion” model [6,8,11]. Therefore, in all soil masses the mechanical wear patterns characteristic of abrasive wear were dominant.

The identified images of the friction surface are reflective of the course of mass wear in the experiment (Figure 6, Figure 7 and Figure 8). The highest values of the wear were noted in the heavy soil and they were three times greater than the wear in the light soil, and 1.5 times greater in the medium soil. TBL Plus steel was characterised by the smallest wear irrespective of the soil type. This relationship is particularly noticeable for heavy soils, where in relation to B27 steel, the wear was almost two times smaller, and for the remaining ones, by approximately 30%. In turn, the highest wear values were noted in the light soil for Hardox 500 steel, in the medium soil for XAR 600 steel and B27 steel in the heavy soil.

## 5. Forecasting the Wear Rate Using the Holm–Archard Model

A comparison of empirical values with the theoretical values determined using the Holm-Archard model is presented in Figure 9, Figure 10, Figure 11, Figure 12, Figure 13 and Figure 14. To better illustrate the obtained relationships, data for only two steels are provided in each figure.

Based on the obtained results, a high correlation coefficient was noted for all materials between the data originating from the model and from experimental test results.

The highest correlation values were obtained for the light and heavy soil, which is confirmed by the results presented in diagrams (Figure 9, Figure 10, Figure 11, Figure 12, Figure 13 and Figure 14). These figures indicate that for B27, Hardox 500 and TBL steels, the data from the model coincide with the experimental testing results.

In the light soil, the model describing the wear for TBL steel, where the difference between values did not exceed 0.000085, was characterised by the best fitting. On the other hand, the model was worst fitted for XAR 600 steel, with differences of 0.00036 being noted. In the medium soil, the highest and lowest coefficients of model fitting were also noted for TBL steel (0.00011) and XAR 600 steel (0.0019), respectively. For the heavy soil, the model was best fitted for Hardox 500 steel (0.0004), while it was the worst fitted for B27 steel (0.0027).

The consistency of the obtained results presenting the overall fitting of the tested materials model in particular soils was verified by applying the coefficient of determination r2 (Table 4, Table 5 and Table 6, Figure 15, Figure 16 and Figure 17).

Based on the results obtained from the theoretical Holm-Archard model and from the experiment, strong correlations between the obtained wear values were proven. The coefficient of determination r2 value obtained using the regression method was 0.97 for the light soil, 0.87 for the medium soil and 0.90 for the heavy soil, which confirms the correctness of the proposed probabilistic model of the wear of wear-resistant steels.

In addition, in order to fit the model to the data obtained from the experiment, Nash–Sutcliff (NS) coefficient was applied [21]:$$NSE=1-\frac{\sum_{t=1}^{T}{\left(Q_{m}^{t}-Q_{o}^{t}\right)}^{2}}{\sum_{t=1}^{T}{\left(Q_{o}^{t}-{\bar{Q}}_{o}\right)}^{2}}$$
where $Q_{o}^{t}$ is the observed value, $\bar{Q_{o}}$ is the mean of observed value and $Q_{m}^{t}$ is the modeled value.

The coefficient is used to assess the fitting of a model to the experimental data. The NS coefficient can take values ranging from −∞ to 1. Value 1 corresponds to the perfect fitting of measurements with the results obtained from the model. The authors assumed that a satisfactory NS coefficient value should fall within the range from 0.80 to 1.

Nash–Sutcliffe coefficient values for particular materials depending on the soil type are presented in Table 7.

The highest NS coefficient value in the light soil was noted for B27 (0.997) and TBL steel (0.995), for which the differences are minor. On the other hand, the lowest coefficient value of 0.964 was noted for Hardox 500 steel. In the medium soil, TBL steel was characterised by the highest value of 0.995, while XAR 600 steel had the lowest value of 0.766. For Hardox 500 steel, the highest NS coefficient value was noted in the heavy soil, while B27 steel was characterised by the lowest value of fitting in this soil.

In accordance with the assumptions adopted by the authors and concerning the fitting of the model values with experimental data except for XAR 600 steel in the medium soil, the NS coefficient had values over 0.80.

Based on the analyses presented above, it can be concluded that the proposed model can be a tool for determining the rate of wear of operating parts made from wear-resistant steels. However, the common use of this model requires further testing on a broader group of materials considering different soil types and operating conditions.

## 6. Conclusions and Discussion

The results obtained from the experimental testing confirmed the possibility for forecasting the mass wear of simple operating parts processing a soil mass using the Holm-Archard model. Mean fitting error for theoretical data ranged from 6.9% in the light soil, through 16.3% in medium soils, to 16.9% in heavy soils. Such a degree of fitting should be regarded as highly satisfactory. The obtained relationships extend the scope of application of the Holm-Archard model during abrasive wear in particular types of abrasive masses.

The authors of this study pointed out that it is impossible to estimate the value of consumption based on derived patterns, because they do not have a numerical solution to the presented equations and a sufficient bank of empirical data. Results of the conducted studies with regard to the wear of operational parts in a soil mass indicate the need for verification of the Holm-Archard model in relation to other measures of wear, construction materials and more complex operational parts. It has been shown in the literature that the usability of these models has been demonstrated inter alia in conditions of micro-crowning, wear with constant abrasive, corrosive and mechanical wear.

Based on the obtained results, it can be concluded that steel for operating parts should be selected in the context of properties of the soil mass subjected to processing. TBL PLUS steel proved to be the least susceptible to a change in the grain-size distribution within the soil mass and, thus, in a way it wears. It is characterised by a structure with post-martensitic orientation with a very few carbide phase precipitates inside the martensite strips. On the other hand, B27 steel, which is also characterised by a structure with post-martensitic orientation with small carbide precipitates inside martensite, is susceptible to their chipping in soils with an intensive effect of fixed sand grains.

The main aim of the study was to verify if the Holm-Archard model, commonly applied under abrasive wear conditions, can also be used for the rather specific tribological pair of steel-soil mass. The obtained results confirmed the possibility of using the Holm-Archard model to forecast the wear of steel in an abrasive soil mass. It should be stressed, however, that in this study, the testing was limited to special steels used to manufacture soil-processing operating parts. They are characterised by a similar chemical composition and a diversified microstructure. The *k* coefficient value reflects the abrasive properties of the soil for a particular type of steel. Complete knowledge of the Holm–Archard model’s suitability for the forecasting of wear in a soil mass can be estimated by performing tests on other materials used to manufacture operating parts, e.g., padding welds.

## Figures and Tables

**Figure 1 materials-12-02180-f001:**
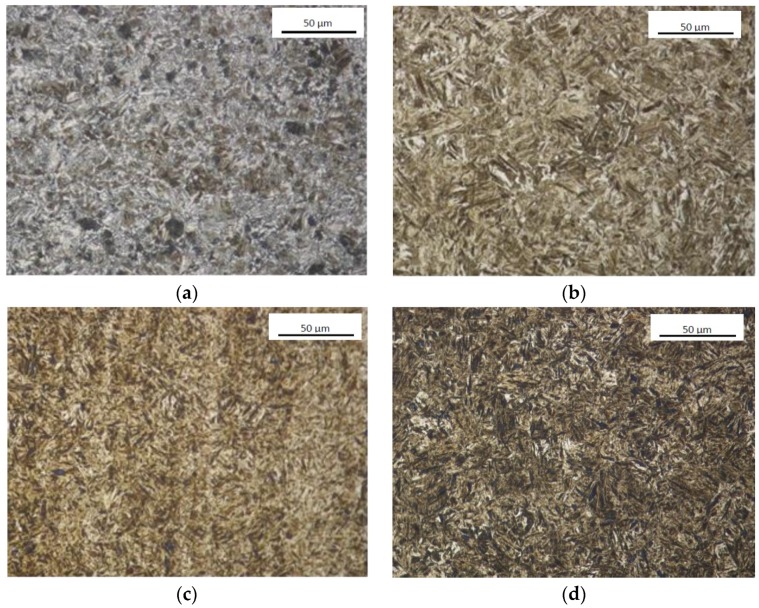
Microstructures of the tested materials. (**a**) B27, (**b**) Hardox 500, (**c**) XAR 600, (**d**) TBL Plus.

**Figure 2 materials-12-02180-f002:**
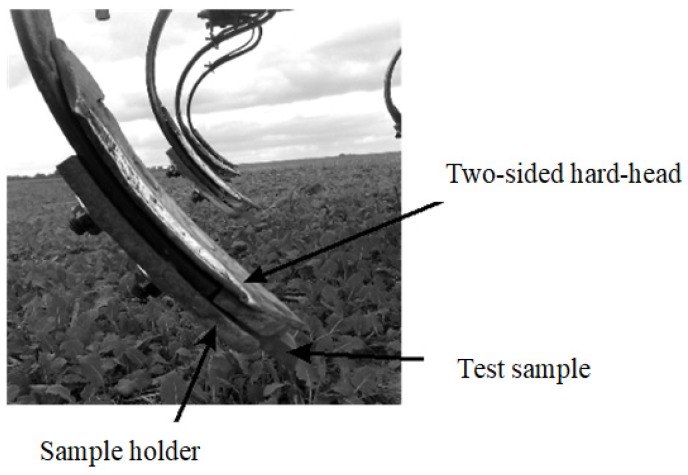
A view of test sample before field testing.

**Figure 3 materials-12-02180-f003:**
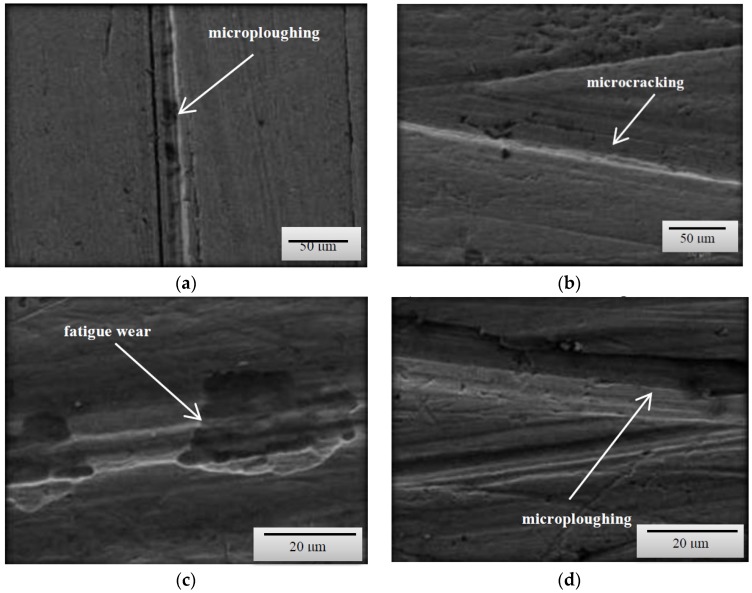
SEM micrographs of the surfaces of tested steels subjected to wear in a light soil (**a**) B27 steel, magnification 50×; (**b**) Hardox 500 steel, magnification 50×; (**c**) XAR 600 steel, magnification 500×; (**d**) TBL Plus steel, magnification 500×.

**Figure 4 materials-12-02180-f004:**
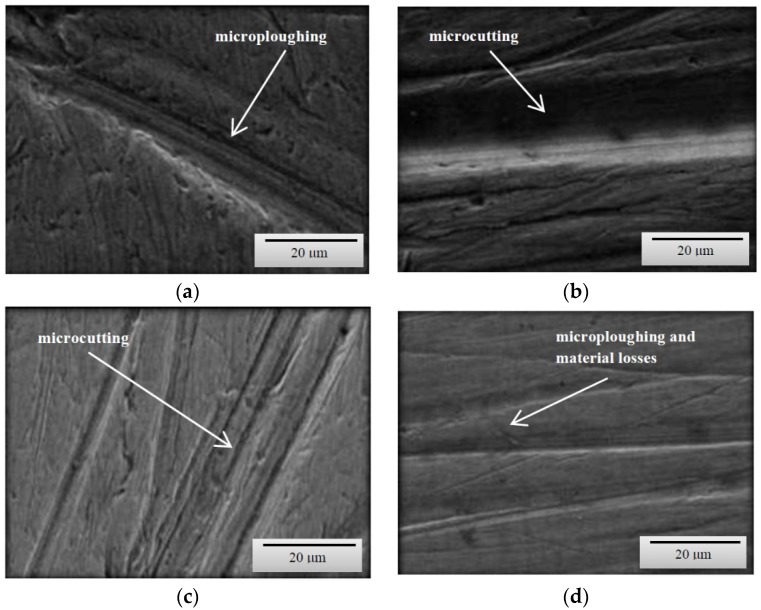
SEM micrographs of the surfaces of tested steels subjected to wear in a medium soil (**a**) B27 steel, magnification 1500×; (**b**) Hardox 500 steel, magnification 1500×; (**c**) XAR 600 steel, magnification 1500×; (**d**) TBL Plus steel, magnification 1500×.

**Figure 5 materials-12-02180-f005:**
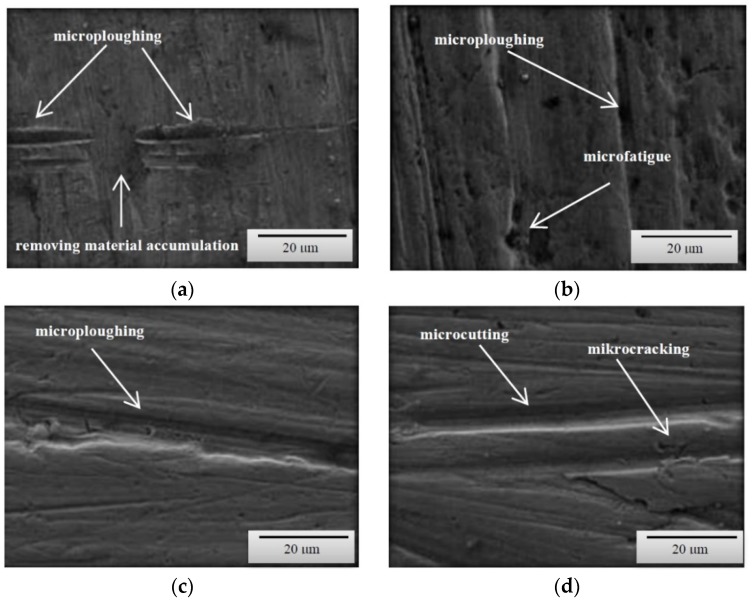
SEM micrographs of the surfaces of tested steels subjected to wear in a heavy soil (**a**) B27 steel, magnification 1500×; (**b**) Hardox 500 steel, magnification 1500×; (**c**) XAR 600 steel, magnification 1500×; (**d**) TBL Plus steel, magnification 1500×.

**Figure 6 materials-12-02180-f006:**
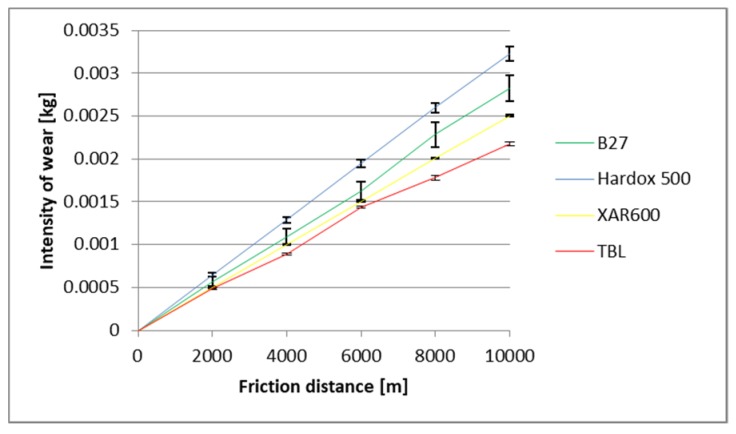
The course of the wear in the light soil.

**Figure 7 materials-12-02180-f007:**
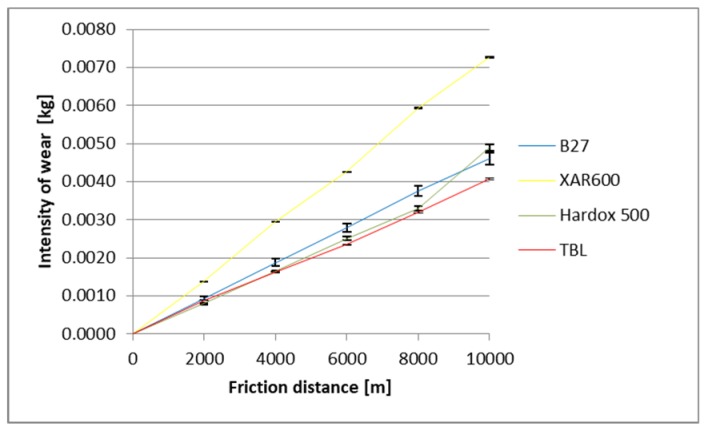
The course of the wear in the medium soil.

**Figure 8 materials-12-02180-f008:**
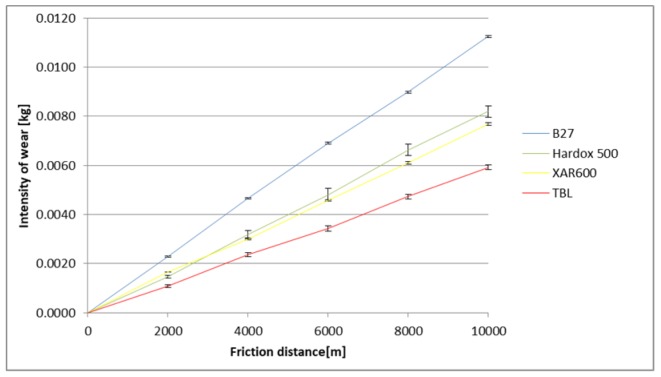
The course of the wear in the heavy soil.

**Figure 9 materials-12-02180-f009:**
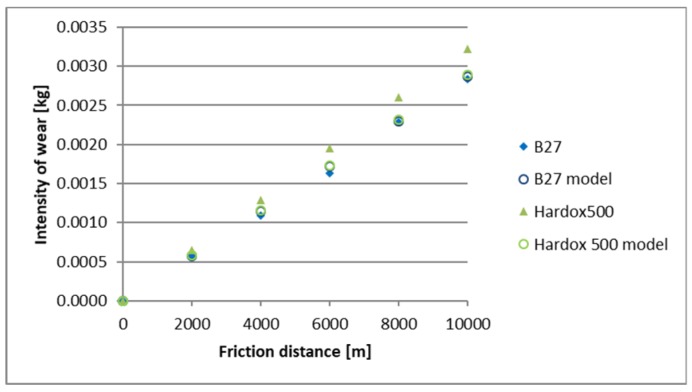
The course of theoretical and empirical wear of B27 and Hardox 500 steels in light soil.

**Figure 10 materials-12-02180-f010:**
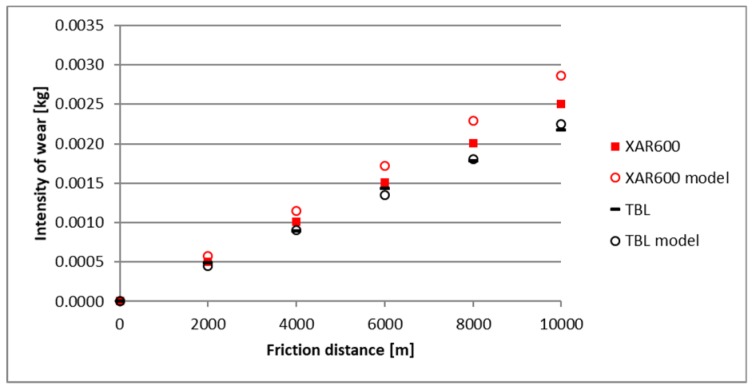
The course of theoretical and empirical wear of XAR 600 and TBL steels in light soil.

**Figure 11 materials-12-02180-f011:**
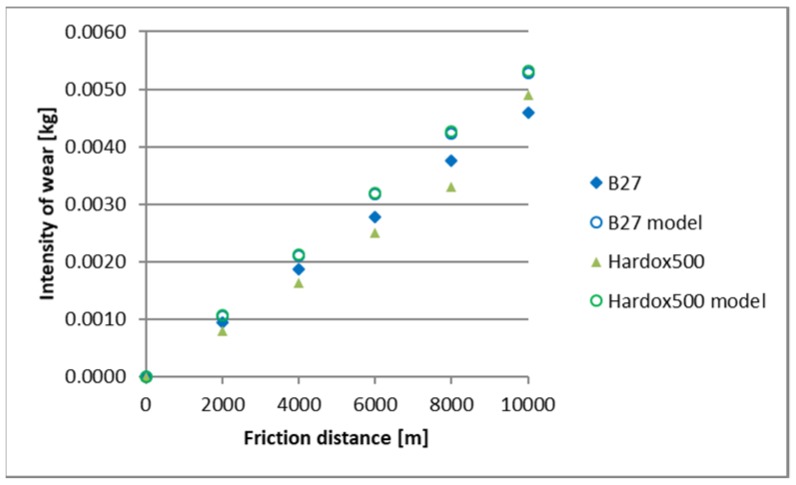
The course of theoretical and empirical wear of B27 and Hardox 500 steels in medium soil.

**Figure 12 materials-12-02180-f012:**
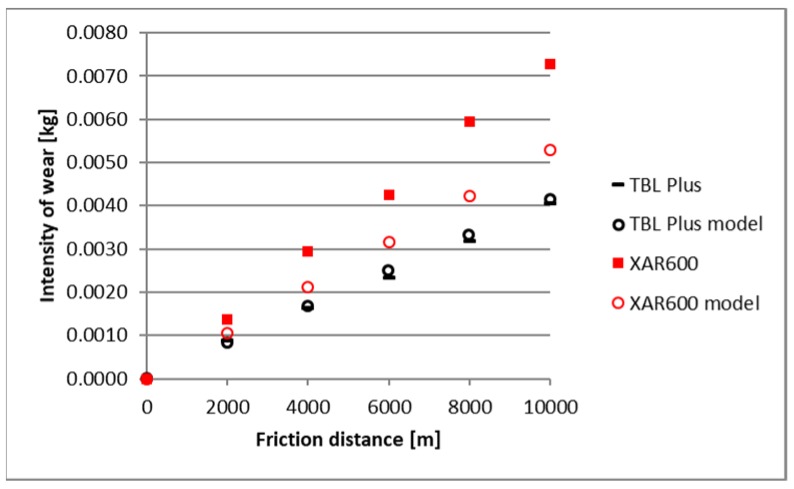
The course of theoretical and empirical wear of XAR 600 and TBL steels in the medium soil.

**Figure 13 materials-12-02180-f013:**
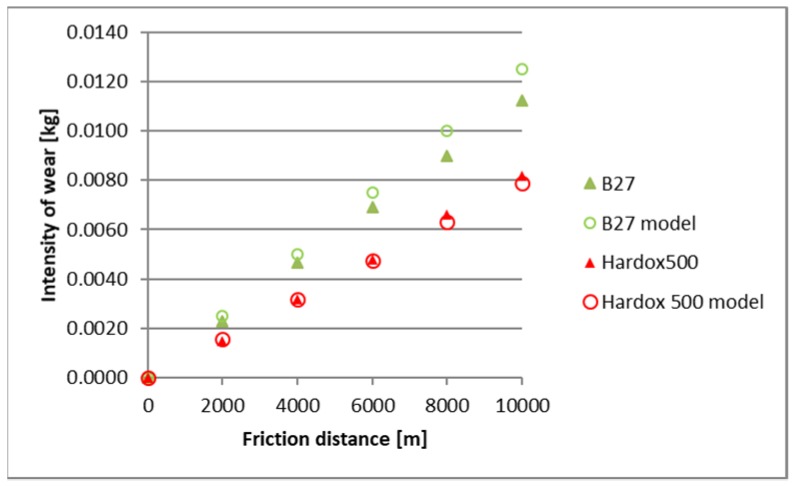
The course of theoretical and empirical wear of B27 and Hardox 500 steels in heavy soil.

**Figure 14 materials-12-02180-f014:**
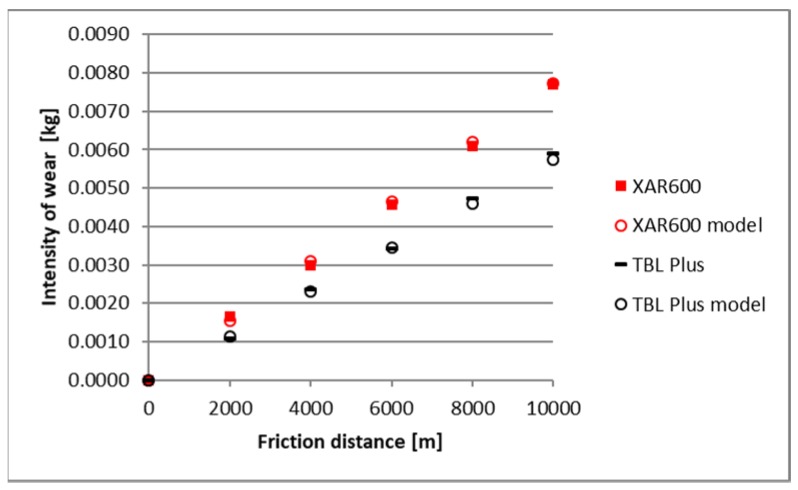
The course of theoretical and empirical wear of XAR 600 and TBL steels in heavy soil.

**Figure 15 materials-12-02180-f015:**
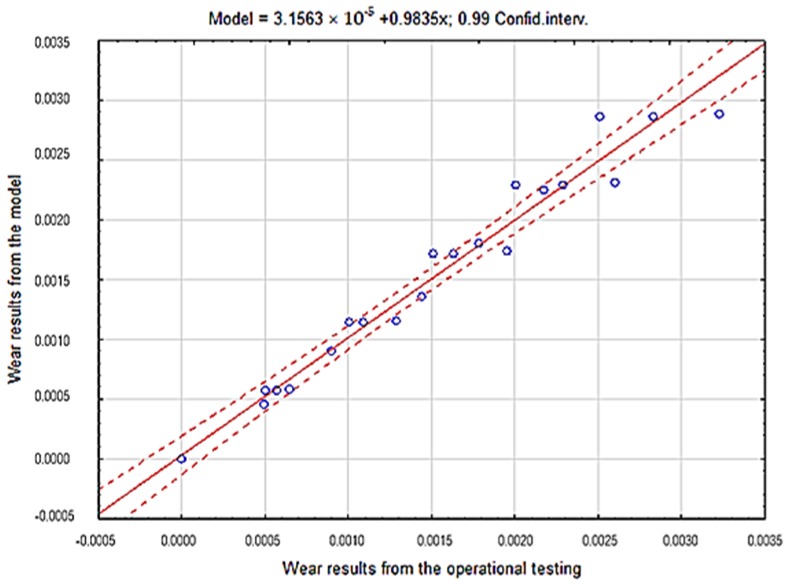
The course of the correlation of empirical data with theoretical data for steel in light soil.

**Figure 16 materials-12-02180-f016:**
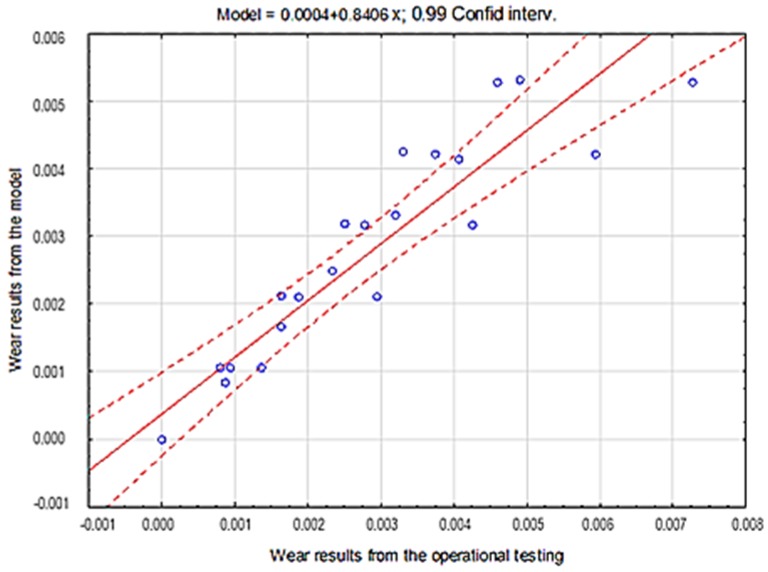
The course of the correlation of theoretical data with empirical data for steel in light soil.

**Figure 17 materials-12-02180-f017:**
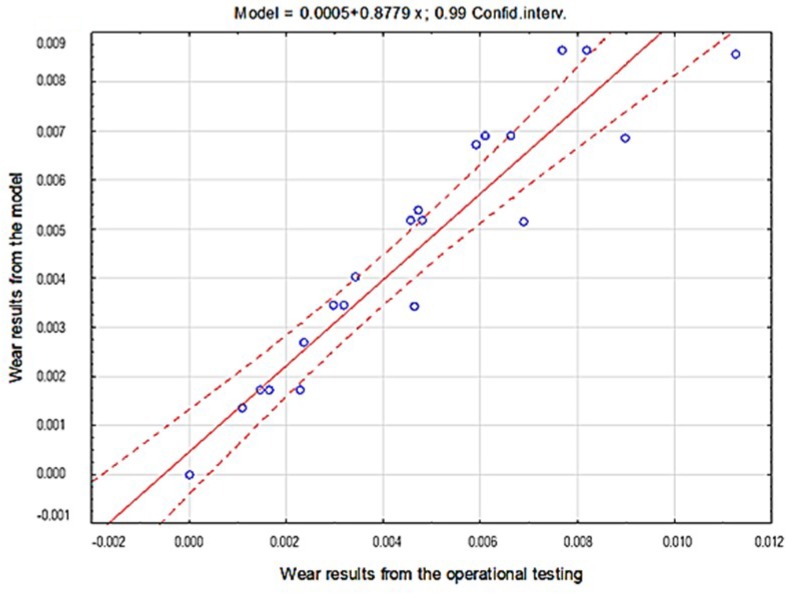
The course of the correlation of theoretical data with empirical data for steel in heavy soil.

**Table 1 materials-12-02180-t001:** Chemical composition of the tested materials.

Elements	B27	Hardox 500	XAR 600	TBL Plus
	Content of Elements (% Weight)			
C	0.28	0.29	0.37	0.34
Mn	1.26	1.05	0.85	1.25
Si	0.23	0.23	0.19	0.21
P	0.009	0.006	0.014	0.012
S	0.006	-	0.001	0.010
Cr	0.32	0.96	0.83	0.25
Ni	0.05	0.05	1.21	0.08
Mo	0.001	0.017	0.15	0.03
V	0.004	0.008	0.002	-
Cu	0.04	0.006	0.03	0.08
Al	0.03	0.055	0.10	0.04
Ti	0.03	0.003	0.003	0.04
Nb	-	-	0.009	-
Co	0.01	0.014	0.005	0.001
B	0.0015	0.0009	0.0021	0.0025

**Table 2 materials-12-02180-t002:** Grain-size distribution of soil.

Soil	Content of Fractions [%]		
	Silt <0.002 (mm)	Dust 0.050–0.002 (mm)	Sand 2.0–0.05 (mm)
Loamy sand	1.69	20.83	77.48
Light loam	7.02	40.32	52.66
Common loam	16.56	49.92	33.62

**Table 3 materials-12-02180-t003:** Data adopted for the model.

Soil Type	Tested Material	k Coefficient Value	P (N)	L (m)	H (N/mm^2^)
Light soil	B27	0.0000028	390	0–10,000	1870
	Hardox 500				1856
	XAR 600				1870
	TBL Plus				2380
Medium soil	B27	0.0000056	550	0–10,000	1870
	Hardox 500				1856
	XAR 600				1870
	TBL Plus				2380
Heavy soil	B27	0.0000088	770	0–10,000	1870
	Hardox 500				1856
	XAR 600				1870
	TBL Plus				2380

**Table 4 materials-12-02180-t004:** Results of correlation analysis for the group of tested steels in light soil.

Variable *X* and Variable *Y*	Correlations Marked with Correlation Coefficients Are Significant at *p* < 0.09000			
	Mean	Standard Deviation	Pearson’s Correlation Coefficient r (X,Y)	Coefficient of Determination r2
Operational	0.00135	0.00096		
Model	0.00136	0.00096	0.9866	0.9734

**Table 5 materials-12-02180-t005:** Results of correlation analysis for the group of tested steels in medium soil.

Variable *X* and Variable *Y*	Correlations Marked with Correlation Coefficients Are Significant at *p* < 0.09000			
	Mean	Standard Deviation	Pearson’s Correlation Coefficient r (X,Y)	Coefficient of Determination r2
Operational	0.0025	0.00197		
Model	0.0025	0.00177	0.9331	0.8706

**Table 6 materials-12-02180-t006:** Results of correlation analysis for the group of tested steels in heavy soil.

Variable *X* and Variable *Y*	Correlations Marked with Correlation Coefficients Are Significant at *p* < 0.09000			
	Mean	Standard Deviation	Pearson’s Correlation Coefficient r (X,Y)	Coefficient of Determination r2
Operational	0.004118	0.003132		
Model	0.004203	0.002979	0.952985	0.908180

**Table 7 materials-12-02180-t007:** Nash-Sutcliffe coefficient results for the tested steels in particular soils.

Materials	Soil		
	Light Soil	Medium Soil	Heavy Soil
B27	0.997	0.939	0.813
XAR 600	0.935	0.766	0.953
Hardox 500	0.964	0.880	0.988
TBL	0.995	0.995	0.932

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
