# Peer review of "Forecasting the Wear of Operating Parts in an Abrasive Soil Mass Using the Holm-Archard Model"

_materials, 2019, doi:10.3390/ma12132180_

Round 1
Reviewer 1 Report
The paper deals with the prediction of wear of steel parts in different types of abrasive soil (light, medium and heavy) by using the Holm-Archard model and comparing it with experimental data. The study shows a very good correlation between the predicted and the measured values for all the 4 steels tested. Generally, the paper is interesting and suitable for publication; however, it contains some major issues in terms of structure and writing. For example, the Introduction section is significantly too long and contains too many information which is not relevant for the present study – it should be shortened to a maximum of half of the current length. Similar is true also for the Conclusions section and there are also some other structure and grammar related issues which should be addressed before re-submitting the paper. A detailed description of requested corrections is provided below.
Correction propositions (provided numbers refer to the line numbers):
- 19, 20: It should be "TBL Plus and B27"
- 22: It should be "for B27"
- Lines 28-97 could be summarized in one paragraph. Lines 108-121 should be omitted since they are not relevant to the present study. The introduction should provide a short overview of the state-of-the-art and goals of the study in that context.
- Relevant information for the introduction begins at line 122; the previous part of the introduction is unnecessary and over-complicated.
- Lines 166-187 do not belong to the section Materials and methods, but rather to the Introduction. However, since the Introduction is already too long, this part could be omitted. Important information can be provided with citations to the relevant literature.
- 197-207: This part can be omitted or significantly shortened. For such information, it is more appropriate to provide citations to the relevant literature.
- 202: New paragraph begins in the middle of the sentence.
- 223-224: Please provide the information on the producer and the model of the hardness tester.
- 230: It should be "optical microscopes were used."
- 258: Please mention that the % proportions refer to the water content.
- 275-278: Please use normal citation style by using numbers in square brackets.
- 279: Reference number is missing.
- Figures 4-6: Description of subfigures (a)-(d) should be provided in the figure captions, not directly in the figures.
- Captions of Figures 4-6: Instead of "Images", the term "SEM micrographs" should be used.
- 297-298: It should be: "The surface damage descriptions depicted in SEM micrographs…"
- 299: Instead of "way of consumption", the term "wear mechanism" should be used.
- 301: Instead of "method of wear", the term "wear mechanism" should be used.
- 312: Instead of "phenomena", the term "mechanism" should be used.
- 313: "occurs" should be at the end of the sentence.
- 325: It should be "a specific adhesive for the hard soil particles".
- 326: Instead of "consumption", the term "wear" should be used.
- Generally, the description of Figures 4-6 is slightly too long (same things are repeated several times) and over-complicated. The description could be shortened and written more clearly/straightforward.
- From the view of the order of results, it would be more logical to initially show and describe the course of wear (Figures 7-9) and afterwards the observed wear mechanisms (Figures 4-6).
- 373-377: Authors claim that in the light soil the model was worst fitted for XAR 600 steel (Figure 11) and in heavy soil for B27 steel (Figure 14); however in Figures 11 and 14, the data for XAR 600 and B27 steels seem to be better fitted than for the other steels (TBL and Hardox 500) in the mentioned diagrams.
- 376: Sentence should end with "respectively", i.e. "steel (0.0019), respectively."
- 393-394: "respectively" should be at the end of the sentence.
- 397: In the reference [22] there is an unnecessary "L".
- 435-450: This part does not belong to the conclusions; it would be better fitted for the Discussion section.
- Generally, the use of the English language is on a sufficient level; however, sometimes sentences are unnecessarily over-complicated and even unclear. If possible, the manuscript should be double-checked by a third reader.
Author Response
The authors thank you for the remarks. Details in the attachment

Reviewer 2 Report
Manuscript No.: Materials-522558
Date received: May 23, 2019
Title: Forecasting the Wear of Operating Parts in an Abrasive Soil Mass Using the Holm-Archard model
Authors: Jerzy Napiórkowski, Magdalena Lemecha, Łukasz Konat
According to the Abstract the paper presents a forecasting of the wear of working elements in an abrasive soil mass using Holm-Archard model. The novelty is that to date, the model has not been verified during the wear in a soil mass, which is a discrete friction surface. Four grades of steel resistant to abrasive wear, intended for the manufacturing of operating parts exposed to wear within a soil mass, Hardox 500, XAR 600, TBL 20 Plus, B27, were subjected to testing.
The authors propose that in this work to assess the suitability of the Holm-Archard model for the forecasting of the wear of operating parts processing an abrasive soil mass.
After carefully reviewing this paper, I recommend that it:
In the line 120, figure 1 is define the Φ{ φ1, φ2, φ3, ..., φτ}– a set of answers which describes the wear process and on the figure appear Φ{ ϕ1, ϕ2, ϕ3, ..., ϕτ} please homogenized the parameters
The introduction is too long. Here are a few examples of Soil Mass Abrasive wear, but it does not provide any experimental results obtained based on the presented models and already existing in literature
Please restructured least the Introduction and provide more experimental data to help validate your results.
In lines 167-173 you introduce “The currently produced steels of this type are characterised by the addition of boron in an amount of 0.002÷0,005% w/w”. In this range of concentrations, boron dissolves in austenite; consequently, even with ordinary volume hardening, a homogeneous structure of fine-grained perlite or martensite with highly fragmented grains over the entire section of the part can be obtained. The currently obtained martensitic structures in low- and medium-carbon steels require no tempering treatment after hardening, and, at the same time, very high rates of static strength and yield point, and a high capacity to absorb dynamic loads can be obtained”.
And in lines 181-187 you introduce “Currently-produced steels of this type are characterised by the micro-addition of boron in an amount of 0.002÷0.005% w/w. Within this range of concentrations, boron dissolves in austenite; consequently, even with ordinary volume hardening, a homogeneous structure of fine-grained perlite or martensite with highly fragmented grains over the entire section of the part can be obtained [18]. Currently-obtained martensitic structures in low- and medium-carbon steels require no tempering treatment after hardening, and, at the same time, very high rates of static strength and yield point, and a high capacity to absorb dynamic loads can be obtained.
The phrase is redundant please give up one of them.
The phrase from line 202 jumped from the middle row on line 203, please take into consideration and rearrange.
In line 215 you talk about HBW hardness and in line 221 about HB, my question is whether we talk about the same Brinell hardness?
And in lines 222 – 223 you introduce “Measurements of the hardness of the tested materials were carried out following the Vickers method”.
In line 228 in Table 3 you introduce the value of the hardness H in N/mm2. Please let me know about correlation with the hardness determined in line 225.
In line 344 referees to the charts from Figures 7, 8, 9 could be more easily understood if on the y-axis would put the unit of measure for wear in grams not in kg, especially as in equation 1 the unit of wear mass measure is also the gram.
I have the same requirement for the charts 10 - 15, more exactly to change from Kg in gram the intensity of wear.
I have a misunderstanding, namely: the coefficient of soil abrasive properties k was obtained on from the relationship 3 based on Holm's and Archard relationship and then you calculated the wear intensity Iz used the same relationship with the one from which the k-coefficient was calculated ... I do not understand the principle, and then you plotted the graphs from figures 10 – 15.
In line 371 you said “the data from the model coincide with the experimental testing results”, it should be if you used the same relationship
In line 379 you introduce the coefficient of determination r2, please explain how you determined this coefficient.
In line 415 you notice “In accordance with the assumptions adopted by the authors and …”, please let me know which are these assumptions.
In lines 391 – 407 you introduce a math model based on Nash-Sutcliff (NS) coefficient. To determine this coefficient you talked about some values Q0 is the mean of observed discharges, and Qm is the modeled discharge. Qmt is the observed discharge at another observation values.
Please let me know for our models which are these parameters.
More elaborate and clear conclusions should be given, including comparisons with the literature.
None of methods can be considered original, nor are the motivations and goals of the experimental efforts very well provided to the reader. Nonetheless, the paper can be of interest for the audience of Materials, mostly because it may represent an additional (compared to the very many similar papers appeared in the last decades).
Author Response

(The authors gave the same response as above.)

Round 2
Reviewer 1 Report
Authors have addressed the issues raised by the reviewer. Only a few minor spelling/grammar mistakes have remained which should be corrected as described below.
Correction propositions (provided numbers refer to the line numbers):
- 179: It should be "were used."
- Captions of Figures 4-6: There is no need to put “-“ after the letters of the subfigures “(a), (b), (c) and (d)”.
- 340: "respectively" at the end of the sentence could be omitted.
Author Response
Thank you for any comments. The correction has been applied.
Reviewer 2 Report
The authors took into account all the suggestions that were made, and completed the entire article.
Nonetheless, the paper can be of interest for the audience of Materials, mostly because it may represent an additional (compared to the very many similar papers appeared in the last decades).
Author Response
Thank you for any comments.